# Assisted Learning: A Framework for Multi-Organization Learning

**Xun Xian**
School of Statistics
University of Minnesota
xian0044@umn.edu

**Xinran Wang**
School of Statistics
University of Minnesota
wang8740@umn.edu

**Jie Ding**
School of Statistics
University of Minnesota
dingj@umn.edu

**Reza Ghanadan**
Google Research
rezaghanadan@google.com

## Abstract

In an increasing number of AI scenarios, collaborations among different organizations or agents (e.g., human and robots, mobile units) are often essential to accomplish an organization-specific mission. However, to avoid leaking useful and possibly proprietary information, organizations typically enforce stringent security constraints on sharing modeling algorithms and data, which significantly limits collaborations. In this work, we introduce the Assisted Learning framework for organizations to assist each other in supervised learning tasks without revealing any organization's algorithm, data, or even task. An organization seeks assistance by broadcasting task-specific but nonsensitive statistics and incorporating others' feedback in one or more iterations to eventually improve its predictive performance. Theoretical and experimental studies, including real-world medical benchmarks, show that Assisted Learning can often achieve near-oracle learning performance as if data and training processes were centralized.

## 1  Introduction

One of the significant characteristics of big data is variety, featuring in a large number of statistical learners, each with personalized data and domain-specific learning tasks. While building an entire database by integrating all the accessible data sources could provide an ideal dataset for learning, sharing heterogeneous data among multiple organizations typically leads to tradeoffs between learning efficiency and data privacy. As the awareness of privacy arises with a decentralized set of organizations (learners), a realistic challenge is to protect learners' privacy concerning sophisticated modeling algorithms and data.

There exists a large market of collaborative learning in, e.g., the internet-of-things [1], autonomous mobility [2], industrial control [3], unmanned aerial vehicles [4], and biological monitoring [5]. We will take the medical industry as an example to illustrate such learning scenarios and their pros and cons. It is common for medical organizations to acquire others' assistance in order to improve clinical care [6], reduce capital costs [7], and accelerate scientific progress [8]. Consider two organizations Alice (a hospital) and Bob (a related clinical laboratory), who collect various features from the same group of people. Now Alice wants to predict Length of Stay (LOS), which is one of the most important driving forces of hospital costs [9].

---

The corresponding website of this project: http://www.assisted-learning.org.

Table 1: Examples of Bob assisting Alice (none of whom will transmit personalized models or data).

| Alice | Hospital | Mobile device | Investor | EEG |
|---|---|---|---|---|
| Bob | Clinical Lab | Cloud service | Financial trader | Eye-movement |
| Collating Index | Patient ID | User ID/email | Timestamp | Subject ID |

**Scenario 1:** If the organization is not capable of modeling or computing, it has to sacrifice data privacy for the assistance of learning. In our example, if Alice does not have much knowledge of machine learning or enough computational resources, she may prefer to be assisted by Bob via Machine-Learning-as-a-Service (MLaaS) [10, 11]. In MLaaS, a service provider, Bob receives predictor-label pairs $(x, y)$ from Alice and then learns a private supervised model. Bob provides prediction services upon Alice's future data after the learning stage by applying the learned model. In this learning scenario, Alice gains information from Bob's modeling, but at the cost of exposing her private data.

**Scenario 2:** If Alice has the capability of modeling and computing, how can she benefit from Bob's relevant but private medical data and model? One way is to privatize Alice's or Bob's raw data by adding noises and transmit them to the other organization. Though this strategy can have privacy guarantees (e.g., those evaluated under the differential privacy framework [12–14]), it often leads to information loss and degraded learning performance. Another solution for Alice is to send data with homomorphic encryption [15]. While it is information lossless, it may suffer from intractable communication and computation costs.

Privacy-sensitive organizations from various industries will not or cannot transmit their personalized models or data. Some common examples are listed in Table 1. Under this limitation, is it possible for Bob to assist Alice in the above two scenarios?

For Scenario 1, Bob could choose to simply receive Alice's ID-label pairs, collate them with his own private data, and learn a supervised model privately. Bob then provides prediction services for Alice, who inquires with future data in the form of an application programming interface (API). For Scenario 2, suppose that Alice also has a private learning model and private data features that can be (partially) collated to Bob's. Is it possible to still leverage the *model* as well as *data* held by Bob? A classical approach for Alice is to perform model selection from her own model and Bob's private model (through Bob's API), and then decide whether to use Bob's service in the future. A related approach is to perform statistical model averaging over the two learners. However, neither approach will significantly outperform the better one of Alice's and Bob's [16, 17]. This is mainly because that model selection or averaging in the above scenario fails to fully utilize all the available data, which is a union of Alice's and Bob's.

Is it possible for Alice to achieve the performance as if all Alice and Bob's private information were centralized (so that the 'oracle performance' can be obtained)? This motivates us to propose the framework of *Assisted Learning*, where the main idea is to treat the predictor $x$ as private and use a suitable choice of '$y$' at each round of assistance so that Alice may benefit from Bob as if she had Bob's data.

The main contributions of this work are threefold. First, we introduce the notion of Assisted Learning, which is naturally suitable for contemporary machine learning markets. Second, in the context of Assisted Learning, we develop two specific protocols so that a service provider can assist others by improving their predictive performance without the need for central coordination. Third, we show that the proposed learning protocol can be applied to a wide range of nonlinear and nonparametric learning tasks, where near-oracle performance can be achieved.

The rest of the paper is organized as follows. First, we briefly discuss the recent development of some related areas in Section 2. Then, a real-world application is presented to demonstrate Assisted Learning's suitability and necessity in Section 3. In Section 4, we formally introduce Assisted Learning and give theoretical analysis. In Section 5, we provide experimental studies on both real and synthetic datasets. We conclude in Section 6.

## 2 Related work

There has been a lot of existing research that considers distributed data with heterogeneous features (also named vertically partitioned/split data) for the purpose of collaborative learning. Early work

on privacy-preserving learning on vertically partitioned data (e.g. [18, 19]) need to disclose certain features' sensitive class distributions. Recent advancement in decentralized learning is Federated Learning [20–23], where the main idea is to learn a joint model using the averaging of locally learned model parameters, so that the training data do not need to be transmitted. The data scenario (i.e., clients hold different features over the same group of subjects) in vertical Federated Learning is similar to that in Assisted Learning. Nevertheless, these two types of learning are fundamentally different. Conceptually, the objective of Federated Learning is to exploit resources of massive edge devices for achieving a global objective. At the same time, the general goal of Assisted Learning is to facilitate multiple participants (possibly with rich resources) to autonomously assist each other's *private* learning tasks. Methodologically, in Federated learning, a central controller orchestrates the learning and the optimization, while Assisted Learning provides a protocol for decentralized organizations to optimize and learn among themselves.

**Vertical Federated Learning**. Particular methods of Vertical Federated Learning include those based on homomorphic encryption, and/or stochastic gradient descent on partial parameters to jointly optimize a global model [24–27]. In all of these schemes, the system needs to be synchronized (among all the participants) in the training process. In contrast, Assisted Learning is model-free, meaning that the models of each participant can be arbitrary. Thus it does not require synchronized updating or the technique of homomorphic encryption.

**Secure Multi-party Computation**. Another related literature is Secure Multi-party Computation [28, 29], where the main idea is to securely compute a function in such a way that no participants can learn anything more than its prescribed output. Several work under this framework studied machine learning on vertically partitioned data [30–32], and they typically rely on an external party. The use of external service may give rise to trust-related issues. Assisted Learning does not require third-party service, and participants often assist each other by playing the roles of both service provider and learner (see Section 4.2).

**Multimodal Data Fusion**. Data Fusion [33–35] aims to aggregate information from multiple modalities to perform predictions. Its focus is to effectively integrate complementary information and eliminate redundant information, often assuming that one learner has already centralized data. In Assisted Learning, the goal is to develop synergies for multiple organizations/learners without data sharing.

## 3 Real-world Applications of Assisted Learning on MIMIC3 Benchmarks

Medical Information Mart for Intensive Care III [36] (MIMIC3) is a comprehensive clinical database that contains de-identified information for 38,597 distinct adult patients admitted to critical care units between 2001 and 2012 at the Beth Israel Deaconess Medical Center in Boston, Massachusetts. Data in MIMIC3 are stored in 26 tables, which can be linked by unique admission identifiers. Each table corresponds to the data 'generated' from a certain source. For example, the *LAB* table consists of patients' laboratory measurements, the *OUTPUT* table includes output information from patients, and the *PRESCRIPTIONS* table contains medications records.

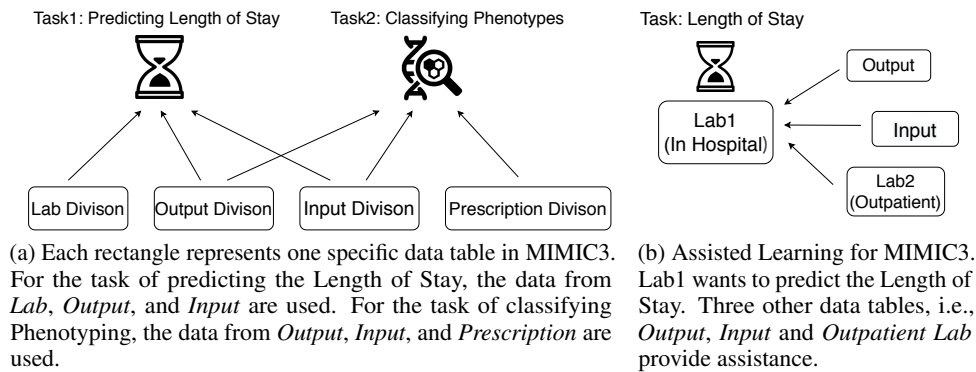

(a) Each rectangle represents one specific data table in MIMIC3. For the task of predicting the Length of Stay, the data from *Lab*, *Output*, and *Input* are used. For the task of classifying Phenotyping, the data from *Output*, *Input*, and *Prescription* are used.

(b) Assisted Learning for MIMIC3. Lab1 wants to predict the Length of Stay. Three other data tables, i.e., *Output*, *Input* and *Outpatient Lab* provide assistance.

Figure 1: Illustrations of (a) MIMIC3 Benchmarks structure and (b) Assisted Learning for MIMIC3 Benchmarks.

MIMIC3 Benchmarks [37, 38] consist of essential medical machine learning tasks, e.g., predicting the Length of Stay (LOS) in order to manage resources and measure patients' acuity and classifying

phenotypes for scientific research. Commonly, every single task involves medical information from different data tables, as depicted in Figure 1a. For example, for the in-hospital Lab (Lab1) to predict LOS, a usual approach is to first centralize data from *Lab*, *Output*, and *Input*, and then construct a model on the joint database. However, sharing sensitive medical data may not be allowed for both data contributors (e.g., patient) and data curator (e.g., hospital), and therefore a method that achieves the best possible learning performance *without* sharing sensitive data is in urgent demand. To summarize, this application domain involves multiple organizations (namely data tables/divisions) with discordant learning goals and heterogeneous/multimodal data whose sharing is prohibited. For Lab 1 to predict LOS via Assisted Learning in the presence of Output, Input, and Lab 2, the main idea is for those learners to assist Lab 1 by iteratively transmitting only task-relevant statistics instead of raw data.

# 4 Assisted Learning

Throughout the paper, we let $X \in \mathcal{X}^{n \times p}$ denote a general data matrix which consists of $n$ items and $p$ features, and $y \in \mathcal{Y}^n$ be a vector of labels (or responses), where $\mathcal{X}, \mathcal{Y} \subseteq \mathbb{R}$. Let $x_i$ denotes the $i$th row of $X$. A supervised function $f$ approximates $x_i \mapsto \mathbb{E}(y_i \mid x_i)$ for a pair of predictor (or feature) $x_i \in \mathcal{X}^p$ and label $y_i \in \mathcal{Y}$. Let $f(X)$ denote an $\mathbb{R}^n$ vector whose $i$th element is $f(x_i)$. We say two matrices or column vectors $A, B$ are *collated* if rows of $A$ and $B$ are aligned with some common index. For example, the index can be timestamp, username, or unique identifier (Table 1).

## 4.1 General Description of Assisted Learning

We first depict how we envision Assisted Learning through a concrete usage scenario based on MIMIC3 Benchmark. Alice (Intensive Care Unit) is equipped with a set of labeled data $(X_0, Y_0)$ and supervised learning algorithms. And $m$ other divisions may be performing different learning tasks with distinct data $(X_i, Y_i)_{i=1,2,...,m}$ and learning models, where $(X_i)_{i=1,2,...,m}$ can be (partially) collated. Alice wishes to be assisted by others to facilitate her own learning while retaining their sensitive information. On the other hand, Alice would also be glad to assist others for potential rewards. A set of learning modules such as Alice constitute a statistical learning market where each module can either provide or receive assistance to facilitate personalized learning goals. In the following, we introduce our notion of algorithm and module in the context of supervised learning. Figure 2 illustrates Assisted Learning from a user's perspective and a service provider's perspective.

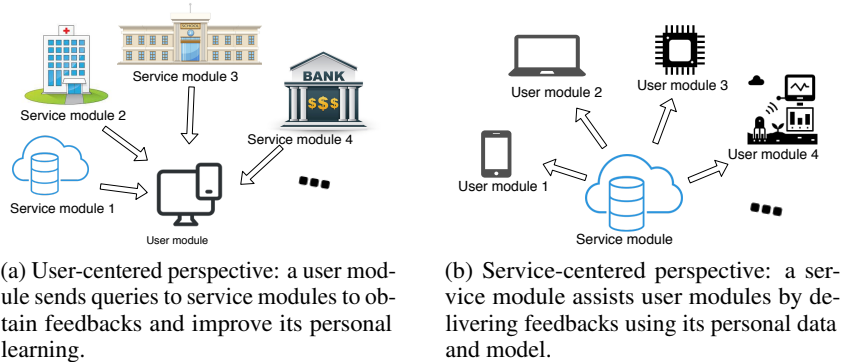

(a) User-centered perspective: a user module sends queries to service modules to obtain feedbacks and improve its personal learning.

(b) Service-centered perspective: a service module assists user modules by delivering feedbacks using its personal data and model.

Figure 2: Assisted Learning from two perspectives.

**Definition 1 (Algorithm).** *A learning algorithm $\mathcal{A}$ is a mapping from a dateset $X \in \mathbb{R}^{n \times p}$ and label vector $y \in \mathbb{R}^n$ to a prediction function $f_{\mathcal{A},X,y} : \mathbb{R}^p \to \mathbb{R}$.*

An algorithm may represent linear regression, ensemble method, neural network, or a set of models from which a suitable one is chosen using model selection techniques [17]. For example, when the least squares method is used to learn the supervised relation between $X$ and $y$, then $f_{\mathcal{A},X,y}$ is a linear operator: $\tilde{x} \mapsto \tilde{x}^{\mathrm{T}}(X^{\mathrm{T}}X)^{-1}X^{\mathrm{T}}y$ for a predictor $\tilde{x} \in \mathbb{R}^p$.

**Definition 2 (Module).** *A module $\mathcal{M} = (\mathcal{A}, X)$ is a pair of algorithm $\mathcal{A}$ and observed dateset $X$. For a given label vector $y \in \mathbb{R}^n$, a module naturally induces a prediction function $f_{\mathcal{A},X,y}$. We simply write $f_{\mathcal{A},X,y}$ as $f_{\mathcal{M},y}$ whenever there is no ambiguity.*

Concerning MIMIC3, a division with its data table and machine learning algorithm is a module. In the context of Assisted Learning, $\mathcal{M} = (\mathcal{A}, X)$ is treated as private and $y$ is public. If $y$ is from a benign user Alice, it represents a particular task of interest. The prediction function $f_{\mathcal{M},y} : \mathcal{X}^p \to \mathcal{Y}$ is thus regarded as a particular model learned by $\mathcal{M}$ (Bob), driven by $y$, in order to provide assistance. Typically $f_{\mathcal{M},y}$ is also treated as private.

**Definition 3 (Assisted Learning System).** *An Assisted Learning system consists of a module $M$, a learning protocol, a prediction protocol, and the following two-stage procedure.*

- *In stage I ('learning protocol'), module $\mathcal{M}$ receives a user's query of a label vector $y \in \mathcal{Y}^n$ that is collated with the rows of $X$; a prediction function $f_{\mathcal{M},y}$ is produced and privately stored; the fitted value $f_{\mathcal{M},y}(X) = [f_{\mathcal{M},y}(x_1), \dots, f_{\mathcal{M},y}(x_n)]^{\mathrm{T}}$ is sent to the user.*

- *In stage II ('prediction protocol'), module $\mathcal{M}$ receives a query of future predictor $\tilde{x}$; its corresponding prediction $\hat{y} = f_{\mathcal{M},y}(\tilde{x})$ is calculated and returned to the user.*

In the above Stage I, the fitted value, $f_{\mathcal{M},y}(X)$, returned from the service module (Bob) upon an inquiry of $y$, is supposed to inform the user module (Alice) of the training error so that Alice can take subsequent actions. Bob's actual predictive performance is reflected in Stage II. The querying user in Stage II may or may not be the same user as in stage I.

### 4.2 A Specific Learning Scenario: Iterative Assistance

Suppose that Alice not only has a specific learning goal (labels) but also has private predictors and algorithm. How could Alice benefit from other modules through the two stages of Assisted Learning? We address this by developing a specific user scenario of Assisted Learning, where we consider regression methods. Note that the scope of Assisted Learning applies to general machine learning problems.

For Alice to receive assistance from other modules, their data should be at least partially collated. For brevity, we assume that the data of all the modules can be collated using non-private indices. Procedure 1 outlines a realization of Assisted Learning between Alice with $m$ other modules. The main idea is to let Alice seek assistance from various other modules through iterative communications where only few key statistics are transmitted. Specifically, the procedure allows each module to only transmit fitted residuals to other modules, iterating until the learning loss is reasonably small.

In the *learning stage* (Stage I), at the $k$th round of assistance, Alice first sends a query to each module $\mathcal{M}_j$ by transmitting its latest statistics $e_{j,k}$; upon receipt of the query, if module $j$ agrees, it treats $e_{j,k}$ as labels and fit a model $\hat{\mathcal{A}}_{j,k}$ (based on the data aligned with such labels); module $j$ then fits residual $\tilde{e}_{j,k}$ and sends it back to module Alice. Alice processes the collected responses $\tilde{e}_{j,k}, \dots$ ($j = 1, \dots, m$), and initializes the $k+1$th round of assistance. After the above procedure stops at an appropriate stopping time $k = K$, the training stage for Alice is suspended. In the *prediction stage* (Stage II), upon arrival of a new feature vector $x^*$, Alice queries the prediction results $\hat{\mathcal{A}}_{j,k}(x^*)$ ($k = 1, 2, \dots, K$) from module $j$, and combines them to form the final prediction $\tilde{y}^*$. Several ways to combine predictions from other modules are discussed in Remark 1. Here, we use unweighted summation as the default method to combine predictions. A simple illustration of the two-stage procedure for Bob to assist Alice is depicted in Figure 3.

It is natural to consider the following oracle performance without any privacy constraint as the limit of learning. Let $\ell$ denote some loss function (e.g. squared loss for regression). The *oracle performance* of learner Alice $\mathcal{M}_0 = (\mathcal{A}_0, X_0)$ in the presence of module $\mathcal{M}_j = (\mathcal{A}_j, X_j)_{j=1,2,\dots,m}$ is defined by $\min_{\mathcal{A}_j} \mathbb{E}\{\ell(y^*, f_j(x^*))\}$ where the prediction function $f_j$ is learned from algorithm $\mathcal{A}_j$ using all the collated data $\bigcup_{i=0}^{m} X_j$. In other words, it is the optimal out-sample loss produced from the candidate methods and the pulled data of all modules. The above quantity provides a theoretical limit on what Assisted Learning can bring to module Alice. Under some conditions, we show that Alice will approach the oracle performance through Assisted Learning with Bob, without direct access to Bob's data $X_B$. In the experimental section, an interesting observation is also presented, which shows a tradeoff between the rounds of assistance and learning performance that strikingly resembles the classical tradeoff between model complexity and learning performance [17].

**Theorem 1.** Suppose that Alice and other $m$ modules use linear regression models. Then for any label $y$, Alice will achieve the oracle performance for sufficiently large rounds of assistance $k$ in Procedure 1.

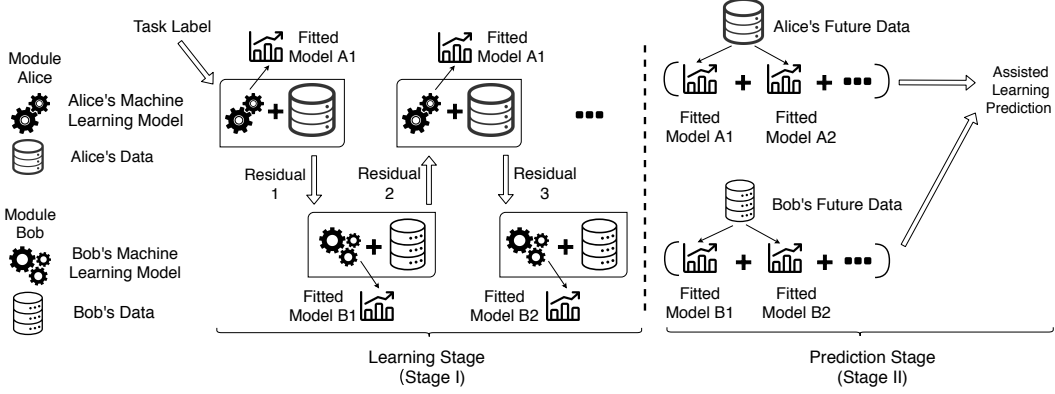

Figure 3: Illustration of the two-stage procedures for Bob to assist Alice.

The above result is applicable to linear models and additive regression models [39] on a linear basis, e.g., spline, wavelet, or polynomial basis. Its proof is included in the supplementary material. The proof actually implies that the prediction loss decays exponentially with the rounds of assistance. The result also indicates that if the true data generating model is $\mathbb{E}(y \mid x) = \beta_a^{\mathrm{T}} x_a + \beta_b^{\mathrm{T}} x_b$, where $x = [x_a, x_b] \in \mathbb{R}^p$ with a fixed $p$, then Alice achieves the optimal rate $O(n^{-1})$ of prediction loss as if Alice correctly specifies the true model.

**Remark 1.** The results can be extended. For example, if $x_a$ and $x_b$ are independent, it can be proved that with one round of communications Alice can approach the oracle model with high probability for large data size $n$; and such an oracle loss approaches zero if $\mathbb{E}(y \mid x)$ can be written as $f_a(x_a) + f_b(x_b)$ for some functions $f_a, f_b$ and if consistent nonparametric algorithms [40, 41] are used. Moreover, suppose that $\mathbb{E}(y \mid x)$ cannot be written as $f_a(x_a) + f_b(x_b)$ but the interactive terms (such as $x_a \cdot x_b$ if both are scalars) involve categorical variables or continuous variables that can be well-approximated by quantizers. The Assisted Learning procedure could be modified so that Alice sends a stratified dataset to Bob which involves only additive regression functions. An illustrating example is $\mathbb{E}(y \mid x) = \beta_a x_a + \beta_b x_b + \beta_{ab} x_{a,1} x_{b,1}$ where $x_{a,1} \in \{0, 1\}$, and Alice sends data $\{x_a : x_{a,1} = 0\}$ and $\{x_a : x_{a,1} = 1\}$ separately to Bob.

### 4.3 Learning with Feedforward Neural Network

In this subsection, we provide an example of how feedforward neural networks can be implemented in the context of Assisted Learning. The data setting will be the same as described in Section 4.1. For simplicity, we consider the learning protocol of Alice and Bob with a three-layer feedforward neural network depicted in Figure 4. Let $w_{a,k}$ (denoted by red solid lines) and $w_{b,k}$ (denoted by blue dash lines) be the input-layer weights at the $k$th round of assistance for Alice and Bob, respectively. Denote the rest of the weights in the neural network at the $k$th round of assistance by $\tilde{w}_k$.

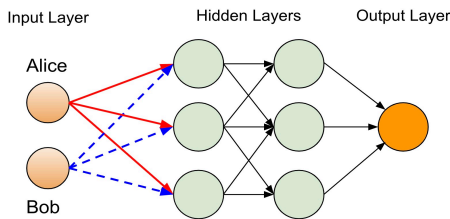

Figure 4: Feedforward neural network via Assisted Learning. For the weights from input to hidden layers, Alice (Bob) can only update her (his) weights denoted by red solid lines (blue dash lines).

In the *learning stage* (Stage I), at the $k$th round of assistance iteration, Alice calculates $w_{a,k}^{\mathrm{T}} X_A$, inquiries Bob's $w_{b,k}^{\mathrm{T}} X_B$, and then combines them to feed the neural network. If $k$ is even, Alice will update her current weight $w_{a,k}$ (resp. $\tilde{w}_k$) to $w_{a,k+1}$ (resp. $\tilde{w}_{k+1}$). Bob will fix his weights for the next iteration, i.e. $w_{b,k+1} = w_{b,k}$. If $k$ is odd, Alice will fix her weights for next iteration, i.e. $w_{a,k+1} = w_{a,k}$, and then sends $\tilde{w}_k$ to Bob. Bob will use the information to update his current weights $w_{b,k}$ to $w_{b,k+1}$. In the *prediction stage* (Stage II), for a new predictor $x^*$, Alice queries the corresponding $w_{b,K}^{\mathrm{T}} x_i^* (i \in \mathcal{S}_b)$ from Bob, and uses the trained neural network to obtain the prediction. The stop criterion we suggest is this: the above procedure of iterative assistance is repeated $K$ times until the cross-validation error of Alice no longer decreases. The above error can be measured by,

e.g., a Stage II prediction using a set of unused data (again, aligned by the data ID of Alice and Bob). The pseudocode is given in Procedure 2. Empirical evaluations on this criterion demonstrate that it typically leads to a near-optimal stopping number. The probability of choosing the optimum could be theoretically derived from large-deviation bounds under certain assumptions.

---

**Procedure 1** Assisted Learning of Module 'Alice' with $m$ other modules (general description)

---

**Input:** Module Alice and its initial label $y \in \mathbb{R}^n$, assisting modules $\mathcal{M}_j$ for $j = 1, 2, \ldots, m$, and (optional) new predictors $\{x_t^*, t \in \mathcal{S}\}$. ($\mathcal{S}$ indexes a set of predictors, and $\mathcal{S}_j$ corresponds to module $\mathcal{M}_j$'s predictors.)
  *Initialization:* $e_{j,k} = y$ $(j = 1, \ldots, m)$, round $k = 1$
1: **repeat**
2:     Alice fits a supervised model using $(e_{j,k}, X_a)$ as labeled data and model $\mathcal{A}_a$.
3:     Alice records its fitted model $\tilde{\mathcal{A}}_{a,j,k}$ and calculates residual $r_{j,k}$.
4:     **for** $j = 1$ to $m$ **do**
5:         Alice sends $r_{j,k}$ to $\mathcal{M}_j$.
6:         $\mathcal{M}_j$ fits a supervised model using $(r_{j,k}, X_j)$ as labeled data and model $\mathcal{A}_j$.
7:         $\mathcal{M}_j$ records its fitted model $\tilde{\mathcal{A}}_{j,k}$ and calculates residual $\tilde{e}_{j,k}$.
8:         $\mathcal{M}_j$ sends $\tilde{e}_{j,k}$ to Alice.
9:     **end for**
10:    Alice initializes the $k + 1$ round by setting $e_{j,k+1} = \tilde{e}_{j,k}$
11: **until** Stop criterion satisfied

---

12: On arrival of a new data $\{x_t^*, t \in \mathcal{S}\}$, Alice queries prediction results produced by the recorded models $\tilde{y}_k = \hat{\mathcal{A}}_{j,k}(x_t^*, t \in \mathcal{S}_j) \in \mathbb{R}^n$, for $j = 1, \ldots, m$ and $k = 1, \ldots, K$.
13: Alice combines (unweighted summation) the above prediction results along with its own records to form a final prediction $\tilde{y}^*$.
**Output:** The *Assisted Learning* prediction $\tilde{y}^*$

---

**Procedure 2** Assisted Learning of Module 'Alice' ('a') using Module 'Bob' ('b') for neural networks

---

**Input:** Module Alice, its initial label $y \in \mathbb{R}^n$, initial weights $w_{a,1}$ (of the input layer) and $\tilde{w}_1$ (of the remaining layer(s)), assisting module Bob, (optional) new predictors $\{x_t^*, t \in \mathcal{S}\}$. ($\mathcal{S}$ indexes a set of predictors, and $\mathcal{S}_a/\mathcal{S}_b$ corresponds to module Alice/Bob's predictors.)
  *Initialization:* round $k = 1$
1: **repeat**
2:     Alice calculates $w_{a,k}^{\mathrm{T}} X_A$ and receives Bob's $w_{b,k}^{\mathrm{T}} X_B$ to train the network in the following way
3:     **if** $k$ is odd **then**
4:         Alice updates $w_{a,k}, \tilde{w}_k$ by using the backpropagation to obtain $w_{a,k+1}, \tilde{w}_{k+1}$, respectively
5:         Bob sets $w_{b,k+1} \leftarrow w_{b,k}$
6:     **else** $\{k$ is even$\}$
7:         Alice sets $w_{a,k+1} \leftarrow w_{a,k}$ and sends $\tilde{w}_k$ to Bob
8:         Bob updates $w_{b,k}, \tilde{w}_k$ by using the backpropagation to obtain $w_{b,k+1}, \tilde{w}_{k+1}$
9:     **end if**
10:    Alice initializes the $k + 1$ round
11: **until** Stop criterion satisfied

---

12: On arrival of a new data $\{x_t^*, t \in \mathcal{S}\}$, Alice calculates $w_{a,K}^{\mathrm{T}} x_t^* (x_t^*, t \in \mathcal{S}_a)$.
13: Alice queries $w_{b,K}^{\mathrm{T}} x_t^*, (x_t^*, t \in \mathcal{S}_b)$ from Bob and combine them to feed the neural network to obtain the final prediction $\tilde{y}^*$.
**Output:** The *Assisted Learning* prediction $\tilde{y}^*$

---

## 4.4 Some Further Discussions

**Related methods**. The idea of sequentially receiving assistance (e.g., residuals), building models, and combining their predictions in Assisted Learning is similar to some popular machine learning techniques such as Boosting [42–46], Stacking [47–49], and ResNet [50]. In Boosting methods, each model/weak learner is built based on the same dataset only with different sample weights. In contrast, Assisted Learning uses side-information from heterogeneous data sources to improve a particular learner's performance. The stacked regression with heterogeneous data is technically similar to one round of Assisted Learning. Nevertheless, a prerequisite for stacking to work in multi-organization learning is that all participants can access labels to train the local ensemble elements. In Assisted Learning, each participant can initiate and contribute to a task *regardless of whether it accesses labels or no*. For example, in the task of MIMIC3 (Sec. 3, Figure 1b), the in-hospital lab aims to predict the Length Of Stay. The outside (outpatient) lab does not have access to in-hospital's private labels, but it could still initiate and assist in the in-hospital lab by fitting the received residuals instead of public labels. The privacy-aware constraint could potentially lead to the failure of stacked regression in multi-organization learning. Also, our experimental studies show that Assisted Learning often significantly outperforms stacking in prediction.

**Adversarial scenarios**. In the previous sections, we supposed that organizations are benign and cooperative during Assisted Learning. In adversarial scenarios, Alice may receive assistance from Bob to steal Bob's local model using queries and responses in the prediction stage. The reconstruction of a trained machine learning model through prediction interfaces is also known as model extraction [51–54]. It is also closely related to the knowledge distillation process [55, 56], which reduces a complex deep neural network to a smaller model with comparable accuracy. Apart from single-model stealing, Alice may also perform multi-model stealing, aiming to learn Bob's capability to generate predictive models for different tasks (built from task-specific labels). If successful, Alice can imitate Bob's functionality and assist other learners as if she were Bob. Such an "imitation" has motivated some new perspective of privacy for Bob's proprietary models/algorithms beyond data privacy (see, e.g., [57] and the references therein) and model protection techniques [54].

## 5 Experimental Study

We provide numerical demonstrations of the proposed methods in Section 4.2 and 4.3. For synthetic data, we replicate 20 times for each method. In each replication, we trained on a dataset with size $10^4$ then tested on a dataset with size $10^5$. We chose a testing size much larger than the training size in order to produce a fair comparison of out-sample predictive performance [17]. For the real data, we trained on 70% of the whole data and tested on the remaining, resampled 20 times. Each line is the mean from replications, and the 'oracle score' is the testing error obtained by the model trained on the pulled data, and the shaded regions describe the corresponding $\pm 1$ standard errors. More examples are included in the supplement.

**Synthetic Data**. We use the data generated by Friedman1 [58], where each data entry contains five features. Suppose that Module A has three features $X_A = [x_1, x_2, x_5]$, Module B has one feature $X_B = [x_3]$, and Module C has one feature $X_C = [x_4]$. In Figure 5(a), with the least square methods, the error terms quickly converge to the oracle. In Figure 5(b), we observe that the performance using nonlinear methods significantly improves on that in (a). In Figure 5(c), with 2-layer neural network models, the error term of Assisted Learning converges slightly slower compared to the oracle, but there is negligible difference regrading the optimal prediction accuracy.

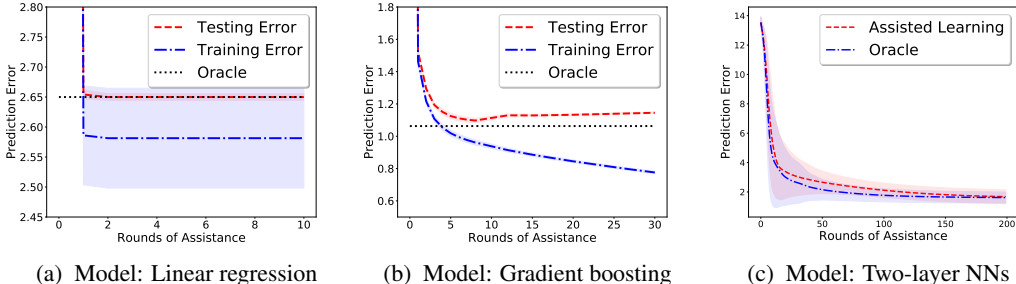

(a) Model: Linear regression    (b) Model: Gradient boosting    (c) Model: Two-layer NNs

Figure 5: Out-sample prediction performance of module A via Assisted Learning (as measured by RMSE) against the rounds of assistance on Friedman1.

**MIMIC3**. We test on the task of predicting Length of Stay, one of the MIMIC3 benchmarks as described in Section 3. Following the processing procedures in [37, 38] with minor modifications, we select 16 medical features and 10000 patient cases. With Module A (Lab table) containing 3 features, module B (ICU charted event table) consisting of 13 features, module B assists module A to predict LOS. We note that the features were naturally split according to data-generating modules, namely hospital divisions. In Figure 6(a) and 6(b), with linear/ridge regression models, the error terms converge to the oracle in one iteration. In Figure 6(c), 6(d), and 6(e), with decision tree and ensemble methods, the testing errors first decrease to the oracle and then begin to increase. Interestingly, this phenomenon resembles the classical tradeoff between underfitting and overfitting due to model complexity [17]. In our case, the rounds of assistance is the counterpart of model complexity. A solution to select an appropriate round was discussed in Section 4.2. In Figure 6(f), with two-layer neural network models, the error terms of Assisted Learning converge slightly slower compared with the oracle, but there is a negligible difference regrading the optimal prediction accuracy.

**Comparison with the stacking method**. We further test the stacked regressions on both Friedman1 and MIMIC3 in previous settings. Various sets of models are chosen for both the base model(s)

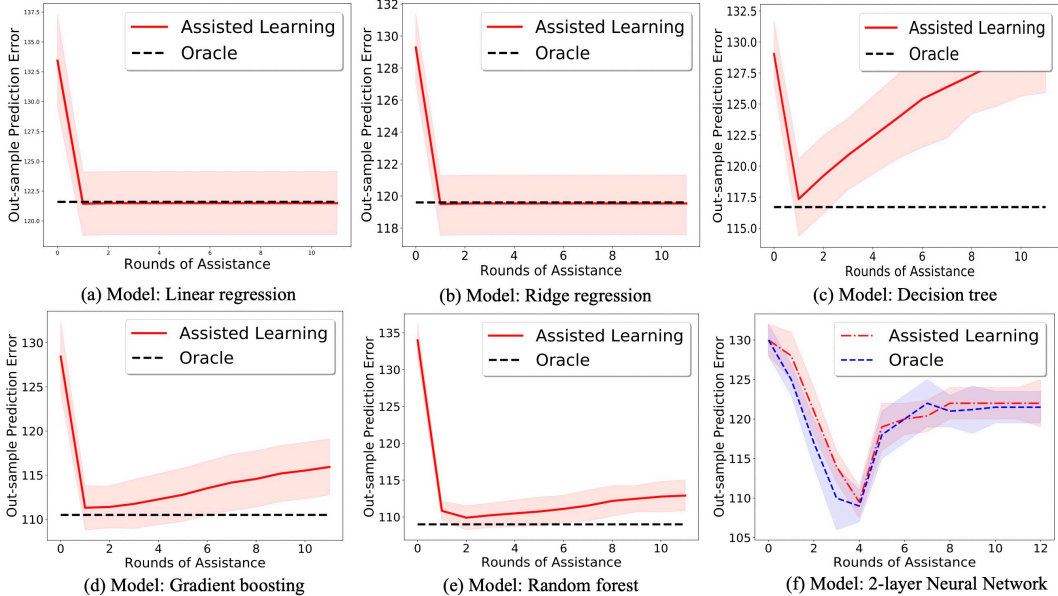

Figure 6: Out-sample prediction performance of module A via Assisted Learning (as measured by mean absolute deviation [37]) against the rounds of assistance on MIMIC3 Benchmark task predicting Length of Stay.

Table 2: Prediction performance of Stacking and Assisted Learning. Column 1 means: in Assisted Learning, participants use LR; in stacking, participants produce features using LR, and the meta-model combining them is LR. Columns 2-4 are similarly defined. Column 5 means: in Assisted Learning, each participant at each round performs model selection to choose from three models; in stacking, each participant produces three features using all the models and ensemble them using RG. Standard errors are within 0.05 over 50 replications.

| Data | Friedman | | | | | MIMIC3 | | | | |
|---|---|---|---|---|---|---|---|---|---|---|
| Base model(s) | LR | RF | GB | GB | LRG | LR | RF | GB | GB | LRG |
| Model for stacking | LR | RF | GB | NN | RG | LR | RF | GB | NN | RG |
| Stacking | 2.63 | 1.68 | 1.42 | 1.39 | 1.35 | 122.7 | 115.8 | 115.9 | 114.2 | 115.9 |
| Assisted Learning | 2.64 | 1.18 | 1.10 | 1.10 | 1.10 | 121.8 | 109.7 | 110.4 | 108.8 | 108.8 |

and the high-level stacking models, including linear regression (LR), random forest (RF), gradient boosting (GB), neural network (NN), ridge regression (RG), and linear regression + random forest + gradient boosting (LRG). From the results summarized in Table 2, Assisted Learning significantly outperforms stacking methods in most cases.

## 6   Conclusion and Further Remarks

The interactions between multiple organizations or learning agents in privacy-aware scenarios pose new challenges that cannot be well addressed by classical statistical learning with a single data modality and algorithmic procedure. In this work, we propose the notion of Assisted Learning, where the general goal is to significantly enhance single-organization learning capabilities with assistance from multiple organizations but without sharing private modeling algorithms or data. Assisted Learning promises a systemic acquisition of a diverse body of information held by decentralized agents. Interesting future work includes the following problems. First, we exchanged the residual for task-specific assistance. What could be other forms of information to exchange for classification and clustering learning scenarios? Second, how does Alice perform statistical testing to decide other organizations' potential utility to initialize Assisted Learning? Third, what is the optimal order of assistance in the presence of a large number of peers?

The supplementary material contains proofs and more experimental studies.

## Broader Impact

The authors envision the following positive ethical and societal consequences. First, the developed concepts, methods, and theories will potentially benefit fields such as engineering, epidemiology, and biology that often involve multi-organizational collaborations since they may not need to share their private models and data. Second, the work will also benefit the general public, whose private data are often held by various organizations. The authors cannot think of a negative ethical or societal consequence of this work.

## Acknowledgement

X. Xian and J. Ding were funded by the U.S. Army Research Office under grant number W911NF-20-1-0222. The authors thank Hamid Krim, Mingyi Hong, Enmao Diao, and Ming Zhong for helpful discussions.

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
