[Supplementary Material · supplementary document for 'Assisted Learning'.pdf]



# Supplement for "Assisted Learning: A Framework for Multi-Organization Learning"

In this supplement, we provide the proof of Theorem 1 and some related discussions. We also provide three additional experiments.

## I. TECHNICAL DETAILS

We will use the following lemma. Let $M$, $N$ be two closed subspaces in a Hilbert Space $\mathcal{H}$. Their Friedrichs angle is defined to be the number $0 \le \theta_F \le \frac{\pi}{2}$ such that

$$\cos \theta_F = \sup_{\substack{\mathbf{x} \in M \cap (M \cap N)^\perp, \mathbf{y} \in N \cap (M \cap N)^\perp \\ \|\mathbf{x}\|, \|\mathbf{y}\| \le 1}} \mathbf{y}^\mathsf{T} \mathbf{x}. \tag{1}$$

**Lemma 1.** [1] Let $M_1, M_2, M_3, \ldots, M_k$ be closed subspaces in $\mathcal{H}$ with intersection $M = \bigcap_{i=1}^k M_i$. For $j = 1, 2, 3 \ldots, k$, we denote $\theta_F^j$ to be the *Friedrichs angle* between $M_j$ and $\bigcap_{i=j+1}^k M_i$. Then, for any $x \in \mathcal{H}$ and any integer $n \ge 1$, we have

$$\|(P_k \ldots P_2 P_1)^n x - P_M x\| \le c^n \|x - P_M x\| \tag{2}$$

where $c = \left(1 - \prod_{j=1}^{k-1}(1 - \cos^2 \theta_F^j)\right)^{1/2}$.

**Proof of Theorem 1.** We prove for the ordinary linear regression of two players (Alice and Bob). The same technique can be extended to general additive models with multiple players. For any design matrix $X \in \mathbb{R}^{n \times p}$, we define the projection matrix $P_X = X(X^\mathsf{T} X)^{-1} X^\mathsf{T}$ and its orthogonal $P_X^\perp = I_n - P_X$. We let $X = [X_A, X_B]$, with $X_A \in \mathbb{R}^{n \times p_1}$, $X_B \in \mathbb{R}^{n \times p_2}$ ($p_1 + p_2 = p$), and $y \in \mathbb{R}^{n \times 1}$ be the corresponding labels. For simplicity, we use $A$, $B$ to denote $span(X_A)$, $span(X_B)$ respectively. Also, we denote $\|\cdot\|$ to be the Euclidean norm and $\|\cdot\|_2$ to be the matrix operator norm.

Denote $e_{\text{orac}}$ to be the residual obtained from the linear regression of $y$ on $X$, i.e., $e_{\text{orac}} = y - \hat{y} = y - (X_A \hat{\beta}_a + X_B \hat{\beta}_b)$, where $[\hat{\beta}_a, \hat{\beta}_b]$ is the oracle least square estimator from all the data. Suppose that Alice holds data $X_A$ and the label $y$, and Bob only has data $X_B$. Let $e_i$ denote the residual at $i$th iteration and $e_0 = y$. Since they both use linear regression models, the residual $e_k$ at $k$th iteration is:

$$e_k = (P_B^\perp P_A^\perp)^k e_0,$$

and we also have the identity

$$e_{\text{orac}} = P_{A \cup B}^\perp e_0 = P_{A^\perp \cap B^\perp} e_0.$$

By Lemma 1, for any integer $k \ge 1$, we have

$$
\begin{aligned}
\|e_k - e_{\text{orac}}\| = \left\|\left(P_B^\perp P_A^\perp\right)^k e_0 - P_{A^\perp \cap B^\perp} e_0\right\| &\le c^k \|e_0 - P_{A^\perp \cap B^\perp} e_0\| \\
&= c^k \|e_0 - e_{\text{orac}}\| = (1 - \sin^2 \theta_F)^{k/2} \|e_0 - e_{\text{orac}}\| \\
&= (\cos \theta_F)^k \|e_0 - e_{\text{orac}}\|, \tag{3}
\end{aligned}
$$

where $\cos \theta_F$ is the *Friedrichs angle* between $A^\perp$ and $B^\perp$. Since $\cos \theta_F = \cos \theta_F^\perp$ [2], and $\cos \theta_F < 1$ (as $X$ has a full column rank), the error term will converge exponentially fast to zero.

In the above arguments, we showed that $e_k$ will converge to $e_{\text{orac}}$ as $k$ become large. Next, we explicitly show the aggregated coefficients obtained by Alice and Bob will asymptotically approach the oracle least square estimators defined above. As a result, Alice will attain near-oracle performance from the assistance of Bob.

**Proposition 1.** Let $\hat{\beta}_a^k$, $\hat{\beta}_b^k$ be the coefficients obtained at the $k$th round of communication for Alice and Bob respectively, and $\hat{\beta}_a$, $\hat{\beta}_b$ be the oracle coefficients (defined as above). Then we have

$$\lim_{k \to \infty} \sum_{i=1}^{k} \hat{\beta}_a^i = \hat{\beta}_a, \quad \lim_{k \to \infty} \sum_{i=1}^{k} \hat{\beta}_b^i = \hat{\beta}_b. \tag{4}$$

**Proof of Proposition 1**. We prove for the case of Alice, i.e., $\lim_{k \to \infty} \sum_{i=1}^{k} \hat{\beta}_a^i = \hat{\beta}_a$. The similar technique can be used to prove Bob's case. From the procedure of Assisted Learning, the $k$th coefficient for Alice is $(X_A^T X_A)^{-1} X_A^T (P_B^\perp P_A^\perp)^{k-1} y$, and we know $\hat{\beta}_a = (X_A^T P_B^\perp X_A)^{-1} X_A^T P_B^\perp y$ by some calculations. Then it suffices to show

$$(X_A^T P_B^\perp X_A)^{-1} X_A^T P_B^\perp = (X_A^T X_A)^{-1} X_A^T \left( \sum_{k=0}^{\infty} (P_B^\perp P_A^\perp)^k \right). \tag{5}$$

By *Gelfand's formula*, we have

$$\rho(P_B^\perp P_A^\perp) \leq \|P_B^\perp P_A^\perp\|_2, \tag{6}$$

where $\rho(\cdot)$ is the spectral radius (the largest absolute value of eigenvalues). From *Spectral Theorem*, we know that for any square matrix $A$, $A$ is normal if and only if the operator norm equals the spectral radii. Therefore, we consider the following two cases.

**Case 1**: If $P_B^\perp P_A^\perp$ is normal, then

$$P_B^\perp P_A^\perp P_B^\perp = P_A^\perp P_B^\perp P_A^\perp \tag{7}$$

We just need to show that

$$X_A^T P_B^\perp = X_A^T P_B^\perp X_A (X_A^T X_A)^{-1} X_A^T \left( \sum_{k=0}^{\infty} (P_B^\perp P_A^\perp)^k \right). \tag{8}$$

Plugging (7) into the right-hand side of (8), we have

$$X_A^T P_B^\perp X_A (X_A^T X_A)^{-1} X_A^T (\sum_{k=0}^{\infty} (P_B^\perp P_A^\perp)^k) = X_A^T P_B^\perp P_A (I_n + P_B^\perp P_A^\perp + P_A^\perp P_B^\perp P_A^\perp + P_A^\perp P_B^\perp P_A^\perp + P_A^\perp P_B^\perp P_A^\perp + \cdots)$$

$$= X_A^T P_B^\perp P_A + X_A^T P_B^\perp P_A P_B^\perp P_A^\perp = X_A^T P_B^\perp P_A + X_A^T P_B^\perp (I_n - P_A^\perp) P_B^\perp P_A^\perp$$

$$= X_A^T P_B^\perp P_A + X_A^T P_B^\perp P_A^\perp + X_A^T P_B^\perp P_A^\perp P_B^\perp P_A^\perp. \tag{9}$$

Since $X_A^T P_B^\perp P_A^\perp P_B^\perp P_A^\perp = X_A^T P_A^\perp P_B^\perp P_A^\perp = (P_A^\perp X_A)^T P_B^\perp P_A^\perp = 0$, then Eq. (9) reduces to

$$X_A^T P_B^\perp P_A + X_A^T P_B^\perp P_A^\perp = X_A^T P_B^\perp (P_A + P_A^\perp) = X_A^T P_B.$$

Therefore, Eq. (8) is correct and Case 1 holds.

**Case 2**: If $P_B^\perp P_A^\perp$ is not normal, then the equality in Eq. (6) will not hold. By a simple fact that $\|P_B^\perp P_A^\perp\|_2 \leq 1$, we have $\rho(P_B^\perp P_A^\perp) < 1$. By the property of *Neumann Series*, the following holds:

$$\sum_{t=0}^{\infty} (P_B^\perp P_A^\perp)^t = (I_n - P_B^\perp P_A^\perp)^{-1},$$

and $(I_n - P_B^\perp P_A^\perp)^{-1}$ exists.

We just need to show

$$(X_A^T X_A)^{-1} X_A^T (I_n - P_B^\perp P_A^\perp)^{-1} = (X_A^T P_B^\perp X_A)^{-1} X_A^T P_B^\perp. \tag{10}$$

By multiplying $(I_n - P_B^\perp P_A^\perp)$ on both sides of (10), it reduces to

$$(X_A^T X_A)^{-1} X_A^T = (X_A^T P_B^\perp X_A)^{-1} X_A^T P_B^\perp - (X_A^T P_B^\perp X_A)^{-1} X_A^T P_B^\perp P_A^\perp.$$

Since $X_A$ is with full column rank, then $X_A X_A^T$ is invertible. Multiplying it on both sides, we have

$$(X_A^T X_A)^{-1} X_A^T X_A X_A^T = (X_A^T P_B^\perp X_A)^{-1} X_A^T P_B^\perp X_A X_A^T - (X_A^T P_B^\perp X_A)^{-1} X_A^T P_B^\perp P_A^\perp X_A X_A^T,$$

which can be verified to be true. Hence, Case 2 holds and we conclude the proof of Proposition 1. In conclusion, if Alice and Bob use linear regression models, then for a sufficiently large number of communications $k$, the oracle performance will be achieved and the error will decay exponentially.

In fact, the Theorem 1 above concerns a finite-sample result when the data size $n$ remains fixed. The following result extends Theorem 1 to a probabilistic setting with random observations and varying $n$. Suppose that the data generating model is $y = \beta_a^{\mathrm{T}} x_a + \beta_b^{\mathrm{T}} x_a + \varepsilon$, where $\varepsilon$ has zero mean and $\sigma^2$ variance, $x = [x_a, x_b] \in \mathbb{R}^p$ follows from a subGaussian distribution with zero mean and correlation matrix $R$, and $x, \varepsilon$ are independent. Suppose that $n$ independent observations $(y_i, x_{a,i})$ are available to Alice, and $(x_{b,i})$ are available to Bob, $i = 1, \ldots, n$. Let $X = [x_1, \ldots, x_n]^{\mathrm{T}}$ denotes the design matrix centralizing all the data. Recall that $\mathcal{S}_a, \mathcal{S}_b$ denote the variable indices of Alice and Bob, respectively.

**Corollary 1.** Assume that the smallest eigenvalue of $X^{\mathrm{T}} X / n$ is almost surely lower bounded by a positive constant. Also assume that $X$ is sub-Gaussian with a fixed covariance matrix, and $\mathbb{E} y^2 < \infty$. Then the final predictor of Alice $\tilde{y}_n^*$ satisfies $\mathbb{E}(\tilde{y}^* - y)^2 \to \sigma^2$ as $n \to \infty$, meaning that it is a consist predictor.

**Proof of Corollary 1.** Let $e_{n,k}$ and $e_{n,\mathrm{orac}}$ denote the residual of Alice at step $k$ of stage I, and the oracle residual by pulling all the data, respectively, where the subscript $n$ highlights their dependence on the sample size. Suppose there are $k$ communications in Stage I. In Stage II, suppose that the aggregated linear prediction function of Alice forms has a coefficient vector $\tilde{\beta}_{n,k}$; also suppose the oracle least square estimate by pulling the data is $\hat{\beta}_n$. It suffices to prove that $\tilde{\beta}_{n,k} - \hat{\beta}_n \to 0$ in probability as $n \to \infty$. By the subGaussian assumption, the *Friedrichs angle* between $X_A$ and $X_B$, $\cos \theta_F$, is bounded away from 1 with probability at most $c_1 p^2 e^{-c_2 n t^2}$ for some constants $c_1, c_2$. Using Theorem 1 and the assumption on the smallest eigenvalue, there exists a constant $c$ that

$$\|\tilde{\beta}_{n,k} - \hat{\beta}_n\|^2 \le c n^{-1} \|X \tilde{\beta}_{n,k} - X \hat{\beta}_n\|^2 = c n^{-1} \|e_{n,k} - e_{n,\mathrm{orac}}\|^2,$$

which goes to zero in probability.

In the above corollary, it is possible that $p \to \infty$ and $k/p \to 0$ as $n \to \infty$, maintaining a high privacy for Bob since only a small fraction of column space is available to Alice.

## II. EXPERIMENTAL STUDY

In this section, we give three additional examples to demonstrate the performance of Assisted Learning.

### A. Synthetic data

We first consider the case where the true data generating function is linear. Let $\mathbf{x}_i^{\mathrm{T}} = [x_{i1}, x_{i2}, \ldots, x_{i6}]$, where $x_{ij} \sim \mathcal{N}(0, 1)$, for $j = 1, 2, \ldots, 6$. The data generating model is $y_i = x_i^{\mathrm{T}} \beta + \varepsilon$, where $\beta \sim \mathcal{N}(\mathbf{0}, I_6)$, and $\varepsilon \sim \mathcal{N}(0, 1)$. Learner A holds features $X_A = [x_1, x_2, x_3]$ and learner B holds features $X_B = [x_4, x_5, x_6]$. The experiments are independently replicated 20 times. Each time a training size of 500 and a testing size of 5000 are used. In Fig. 1(a), with kernel ridge regression method, the error terms quickly converge to the oracle. In Fig. 1(b), we observe that the testing errors first decrease to the oracle scores and then begin to increase. In Fig. 1(c), with 2-layer neural network models, the error term of Assisted Learning converges slightly slower compared to the oracle, but there is negligible difference regarding the optimal prediction accuracy.

*Further discussions on pathetic scenarios*. In practice, Bob may not provide utility because his data is adversarial, irrelevant, or mis-aligned. We discussed adversarial scenarios in Subsection 4.4 of the main paper. We experimented cases where Bob's data is not relevant (e.g., pure noise or a subset of Alice's data). In those cases, Alice will observe unimproved or even degraded performance in the learning and prediction stages. Also, our experiments assumed that participants can correctly collate their data according to a data identifier. It would be interesting future work to study the robustness against a small portion of misalignment. Overall, a practical guide for Alice to prevent pathetic scenarios is to reserve a small portion of data for continuous validation of the utility offered by others.

### B. Real data: Superconductor

We demonstrate our approach using the *Superconductor Data* [3] that consists of 21263 entries and 81 features. The learning task is to use chemical characteristics to predict the superconducting critical temperatures. The features

(a) Model: Kernel ridge  (b) Model: Gradient boosting  (c) Model: Two-layer Neural Network

Fig. 1: Prediction performance for Module A (as measured by RMSE) on linear data with three learning algorithms. Each line is the mean from independent replications, and the shaded regions represent +1/-1 standard errors.

are partitioned into two sets, 40 features held by module A and the other 41 features held by B. We consider two settings where one module uses gradient boosting and the other one uses linear regression.

The results as depicted in Fig. 2(a), 2(b) and 2(c) show that module A can achieve the oracle with negligible differences. The prediction performance for gradient boosting is much better than linear regression. In terms of convergence rate, gradient boosting converges faster than linear regression. In Fig. 2(b), we again observe the 'over-fitting' as the round of assistance increases.

(a) Model: Linear regression  (b) Model: Gradient boosting  (c) Model: Two-layer Neural Network

Fig. 2: Prediction performance for Module A (as measured by RMSE) on Superconductor Data with three learning algorithms. Each line is the mean from independent replications and the shaded regions describe +1/-1 standard errors.

### C. Model Diversity to enhance Assisted Learning

The purpose of this experiment is to demonstrate that Assisted Learning can flexibly allow each organization/learner to bring its unique modeling algorithm to enhance the learning experience. In other words, the assistance is achieved not only by complementary data information but also diverse model advantages.

Suppose that 200 samples are generated from $y_i = X_i^{\mathrm{T}}\beta + \varepsilon$, with $\varepsilon \sim$ *Cauchy(0,1)*, $\beta = [\beta_1, \beta_2, ..., \beta_{50}]$, with $\beta_k \sim \mathcal{N}(0,1)$ for $k = 1, 2, ..., 12$ and $\beta_k = 0$ for $k = 13, ..., 50$, $X_i^{\mathrm{T}} = [x_{i1}, x_{i2}, x_{i3}, x_{i4}, x_{i5}, ..., x_{i50}]$, $x_{ij} \sim \mathcal{N}(0,1)$ for $j = 1, 2, 3, ..., 50$. Module A holds the first 6 features $[x_1, x_2, ..., x_6]$, and module B holds the remaining features. Suppose that B assists A where each of them uses a linear model. Because of the high dimensionality in B and

Fig. 3: Out-sample prediction performance of Module A under three scenarios of Assisted Learning, showing the potential advantages brought by diverse modeling algorithms.

many potential outliers in A and B (due to heavy-tailed noises), the ultimate out-sample performance of A can be unsatisfactory even under assistance. This is shown by the blue line in Figure 3. If module B employs a penalized regression method, e.g., Lasso [4], then the prediction performance will be improved as shown by the green line. Moreover, if module A employs robust techniques such as Huber Regression (HR) [5], then the performance will be further improved shown by the red line.