[Reviews · NeurIPS 2020]

Review 1

Summary and Contributions: The paper proposes an interesting framework for multiple organization learning, where multiple collaborators can aid supervised learning without sharing private algorithms or data. The approach is implemented via iterative transmissions of task specific statistics (e.g., model residuals) between the collaborators, which leverage side information from alternative models to improve learning. Theoretical analyses shows that the method can achieve lossless learning performance in linear regression models. The approach is validated in synthetic and real data experiments, where the proposed method is compared to an oracle performance achieved by models trained with all the combined data.

Strengths: The proposed approach is interesting, and the paper is very well written and organized. The paper does a very good job motivating the need for the approach, and describes with concrete examples, how it can be used to fulfill this need. The paper does a good job describing its novel contributions relative to previous related work in the literature. The paper also provides theoretical analyses that show that the approach is lossless for linear regression models. The paper illustrates the application of the method using both synthetic and real data sets using linear models (linear and ridge-regression), tree based models (decision trees, gradient boosting, and random forests), and neural networks.

Weaknesses: The paper states that model selection or model averaging approaches will not significantly improve over the best of the models (Alice’s or Bob’s) used in the assisted learning procedure because they fail to utilize the full data (the union of Alice’s and Bob’s features). However, ensemble techniques such as stacked regression (Breiman 1996) are often successfully used to improve predictive performance by combining not only different models trained on the same set of features, but also by combining different models trained on different subsets of features. In all experiments performed in the paper, only comparisons between assisted learning and the oracle model were presented. The paper would be considerably stronger if it was able to show that assisted learning compared favorably against (for instance) a stacked model generated with the predictions obtained from the different models on modules M_1, …, M_m (trained with the original public responses). Note that under the assumptions made by the paper, that the labels/response (as well as, some sort of identifier needed to collate the labels/response to the features) are public available, a simpler ensemble approach (such as stacking) could also be directly used to improve learning without sharing the private feature data. In other words, such an approach might serve the same goals of assisted learning without requiring the iterative transmissions of model residuals (as described in Procedure 1) or of predictions from the first layer of neural nets (as described in Procedure 2) to improve predictive performance. While the paper is very interesting, it would be considerably more appealing, if it provided comparisons and empirical evidence that assisted learning can outperform these simpler approaches. Also, the paper does not describe how the predictions generated by the different modules are combined to form the final predictions. How exactly, the predictions are combined in step 13 of Procedure 1? Is simple unweighted averaging used? Or, are more sophisticated approaches used?

Correctness: The approach appears to be technically sound.

Clarity: The paper is very well written and easy to follow. The approach is very well motivated and most of the time described in detail with plenty illustrative examples (although some important details, such as how the predictions are combined into a final prediction, are missing).

Relation to Prior Work: The paper provides a good overview of related work, clearly discussing the relationships between the proposed method and previous related work.

Reproducibility: Yes

Additional Feedback: Thank you for the thorough response (especially, the additional experiments comparing against an ensemble method). I feel the paper is much stronger now by showing that assisted learning can often outperform stacking. Also, thanks for clarifying that each participant can initiate and contribute to assisted learning regardless of whether the participant has access to the labels or not by simply providing residuals instead of public labels. Sorry, I missed this obvious point … The authors should emphasize that assisted learning is more broadly applicable than ensemble methods which require the labels. This is an important practical advantage of the method. Thanks, as well, for the clarifications concerning the way you combined the predictions. I have adjusted the overall score up to an accept. ################################################## As pointed above, the paper would benefit from an empirical comparison against simpler ensemble methods that could still be used under the assumed conditions (that the labels/response, as well as, an identifier used to collate the labels/response to the distinct features from multiple modules, are public available). The paper should compare the performance of the proposed method against at least one of these alternative approaches (stacking would be one such candidate). Also, the paper would benefit from a more nuanced discussion around the key assumption that the response/label data is publicly available (as well as, the assumption that the identifier needed to collate the labels/response to the features is also available). It is reasonable to expect that these assumptions will not hold in many applications. (For instance, in health applications responses/labels might often represent protected information.) The paper would benefit from further discussions and examples (in real world settings) where these assumptions hold. Minor comments: Line 111: “output information patients” => “output information from patients” Line 229: “Denote the rest weights” => “Denote the rest of the weights” Line 257 (also caption of Figure 5): “Friedman1” => “Friedman” Supplement (page 1, last paragraph): “show the the aggregated” => “show the aggregated” and “Alice ad Bob” => “Alice and Bob” Supplement (proof of proposition 1): “We proof for the case” => “We prove for the case” Supplement (A. Synthetic data): “difference regrading the” => “difference regarding the” Supplement (B. Real data: Superconductor): “we obverse the” => “we observe the”


Review 2

Summary and Contributions: The paper introduces assisted learning, which is a framework for enabling to make use of sensitive data from other entities / organizations without sharing raw data. The gist of the approach is that entities learn models for a specific target variable based on their own set of features and only share their error residuals with other entities. The authors therefore stipulate a novel learning protocol where involved entities sequentially exchange novel models which are fitted to the error residuals. The framework is evaluated on synthetic- and medical data (from the MIMIC3 benchmark ‘length of stay’), where the authors show that adequate prediction errors can be achieved in the novel, decentralized learning setting.

Strengths: * Novel, potentially impactful learning framework for multi-organization ML. The approach is sufficiently novel and relevant. * Adequate evaluation showing the feasibility of the approach with distributed feature subsets * The presentation of the work is widely-understandable, providing a sound introduction to technicalities.

Weaknesses: * For the approach to work, data needs to be aligned, i.e. approach is constrained to organizations which hold data about the same instances * While adequate, the evaluation is small-sized with respect to tested data sets

Correctness: The proposed method is sound, as the stipulated claims are backed with proofs and the empirical evaluation follows standard practice

Clarity: The paper, in general, is well-written in that it thoroughly motivates the novel framework, gives illustrative example scenarios and provides formalisms and proofs only when required. Section 4 is, however, a tad hard to read, although this could be easily remedied if procedure 1 is shown earlier. The empirical evaluation is rather small-sized, but covers realistic datasets (e.g. MIMIC3).

Relation to Prior Work: The related work comprises vertical federated learning, secure multi-party computation and multimodal data fusion. Albeit high-level, it provides a sufficient overview of related frameworks, such that the reader can understand the different interaction scheme and learning goal.

Reproducibility: Yes

Additional Feedback: The employed learning approach withing the decentralized protocol seems strongly related to gradient boosting if I am not mistaken. It would be interesting to show the relationship. Minor comments: * The set $S$ not introduced before (in procedure 1) * There might be an indice error: a module is defined as $M_j$, then you use round $k$ and finally refer to module $k$ (page 5) ### Update after author response ### I want to thank the authors for their answers. After having read the rebuttal as well as the other reviews, I remain with my positive score for the paper, as all questions of the reviewers have been thoroughly addressed. I find the proposed assisted learning framework to be original and well-executed.


Review 3

Summary and Contributions: This paper proposed Assisted Learning framework to tackle the challenge between data privacy and learning efficiency. Authors developed three algorithms: Linear regression, Gradient boosting, Two-layer NNs. Experiment results indicates the efficiency of the proposed method.

Strengths: The paper introduce the notion of Assisted Learning to tackle the challenge between data privacy and learning efficiency without sharing data. Moreover, in the context of Assisted Learning, we develop two concrete protocols so that a service provider can assist others by improving their predictive performance. The paper is overall well-presented and easy to understand.

Weaknesses: First, in some context, the ideal is similar to Residual network, but authors did not cite the related works. Second, How to evaluate correctness of the stop criterion? Third, experiments are insufficient. Authors should compare with more related algorithms.

Correctness: Yes

Clarity: Good

Relation to Prior Work: Yes

Reproducibility: Yes

Additional Feedback:


Review 4

Summary and Contributions: The paper presents a method of assisted learning where multiple parties share relevant metrics iteratively in order to train one particular party at a certain task. The contributions are as follows: 1. This work introduces the problem of Assisted Learning. 2. The work discusses two methods of solving this problem.

Strengths: Soundness of the claims: The claims in the paper have been demonstrated by the experiments. Where possible, claims have also been proven theoretically. Significance and novelty: The posed problem certainly appears to be novel. This work can be quite significant when multiple agencies want to improve their predictions without compromising their data or proprietary methods. Relevance: The work is quite relevant to NeurIPS and the ML community in general.

Weaknesses: Soundness of the claim: Although the results were demonstrate on both simulated and real data sets, it was not clear how features were split among the various modules. Was the split random or was it curated by the authors to achieve the best possible results?

Correctness: Yes, claims and method are correct. Empirical methodology can be improved by considering datasets with large number of features. Real world datasets usually have hundred of features. A lot of agencies might share many features. For example, financial firms most often use credit bureau data. So, two or more financial firms are likely to use very similar data. It isn't clear how the performance will get affected as we have increasing number of similar or linearly dependent features across different modules. Perhaps future work should address that.

Clarity: Yes, the paper is well written and easy to follow.

Relation to Prior Work: The authors clearly distinguishes their work with prior work and related fields, like vertical federated learning.

Reproducibility: No

Additional Feedback: The authors don't discuss how features were split across the modules in the experiments. Reporting which feature was used for which module would help make the results reproducible. It might help if the authors could discuss some (imaginary but) pathetic scenarios like: 1. Alice and Bob have the same copy of data and use the same ML algorithms. In this case, Oracle and Alice should always be at par. 2. What if Bob uses a random label generator as a preedictor? 3. What if the data that Bob has is completely irrelevant to the predictive task that Alice has? I have read authors' feedback and stand by my review.

[Author Response · NeurIPS 2020]

We sincerely thank all the reviewers for their encouraging and constructive comments, especially during a difficult time. We are pleased that they found the paper well written and acknowledged the novelty and potential significance of the proposed learning framework as "... potentially impactful learning framework for multi-organization ML" (R2) and "can be quite significant when multiple agencies ..." (R4). We will incorporate all the comments into the revised paper.

**Reviewer#1** *Assumptions on label/response and identifier.* We will clarify in the revision that in assisted learning, each participant can initiate and contribute to a task regardless of whether it accesses labels or not. For example, in the task of MIMIC3 (Sec. 3, Fig. 1b), the in-hospital lab aims to predict Length Of Stay. The out-side lab doesn't have access to in-hospital's private labels, but it could still initiate and provide assistance to the in-hospital lab by fitting the received residuals instead of public labels. In contrast, a prerequisite for ensemble methods to work in multi-organization learning is that all participants (including the outside laboratory) have access to labels in order to train ensemble elements. As suggested, we will discuss more on the assumption of the identifiers needed to collate data.
*Evidence that assisted learning can outperform stacking.* Assuming that labels are public to each participant, we found that Reviewer 1's suggestion on comparing assisted learning and ensemble approaches such as stacking theoretically and practically intriguing. In some cases, we were able to prove that assisted learning significantly outperforms stacking. An example case is where each participant holds a disjoint subset of the features and uses linear regression. The linear coefficients of Alice's features in a stacking model would be determined by her linear space, which may not be proportional to the linear coefficients of Alice's features in an oracle model determined by all the participants' joint linear space. We also did extensive experiments in the last few days to provide empirical evidence (summarized in **Table 1**). The results show that assisted learning often outperforms stacking (under the same settings as in the paper).
*Ways for combining predictions.* We used unweighted sum in Procedure 1. We briefly discussed some sophisticated extensions on page 6, e.g., to exchange and summarize information on stratified data (using categorical variables). We plan to publish a website containing open-source software APIs for a list of case studies and extensions for the work.

**Table 1:** Prediction performances of stacking and assisted learning (LR: linear regression, RF: random forest, GB: gradient boosting, NN: neural network, and RG: ridge regression). The settings are the same as in the paper. Column 1 means: in assisted learning, participants use LR; in stacking, participants produce features using LR, and the meta-model combining them is LR. Columns 2-4 were similarly defined. Column 5 means: in assisted learning, each participant at each round performs model selection to choose from three models; in stacking, each participant produces three features using all the models and ensembled using RG. Standard errors are within 0.05 over 50 replications.

| Data | Friedman | | | | | MIMIC3 | | | | |
|---|---|---|---|---|---|---|---|---|---|---|
| Base model(s) | LR | RF | GB | GB | LR+RF+GB | LR | RF | GB | GB | LR+RF+GB |
| Model for stacking | LR | RF | GB | NN | RG | LR | RF | GB | NN | RG |
| Stacking | 2.63 | 1.80 | 1.76 | 1.68 | 1.60 | 121.7 | 118.1 | 119.3 | 119.7 | 115.9 |
| Assisted learning | 2.64 | 1.31 | 1.23 | 1.23 | 1.25 | 120.5 | 109.7 | 111.3 | 111.3 | 110.8 |

**Reviewer#2** *Alignment of data & Evaluation size.* Each participant needs to hold and distribute identifiers for data items, so that the data from different participants can be conceptually combined. Interesting future work includes assisted learning under partially-aligned identifiers (e.g. timestamps), and robustness against a portion of misalignment. We are developing open-source APIs for large-scale deployment on cloud platforms. Upon the acceptance of this work, we will provide assisted learning services to numerous organizations and further evaluate the framework at larger scales.
*Relationship with gradient boosting.* The process of sequentially building models and combining their predictions in assisted learning is similar to that in Boosting methods. However, in Boosting, each model is built based on the same dataset with different sample weights. In contrast, assisted learning uses side-information from heterogeneous data sources to improve the performance of a particular learner. The relationship will be discussed in our revision.

**Reviewer#3** *Related work.* We will cite ResNet and more related work such as ensemble methods in the revision.
*Evaluation of stop criterion.* We evaluated the proposed stop criterion in all the experimental studies, and we find that the current criterion (based on cross-validation) leads to a near-optimal stopping number. The probability of choosing the optimum could be theoretically derived from large-deviation bounds (under certain assumptions). Future work includes the study of time-dependent data (so backtesting has to be considered). We will discuss these in the revision.
*More experiments.* We will include more comparisons with stacking methods (preliminary results in Table 1).

**Reviewer#4** *Feature splitting.* Feature splitting in the paper is categorized into two types. For real data such as the MIMIC3, the data was naturally split according to data-generating modules (namely hospital divisions). For real data without knowing the data-generating process (Superconductor in Appendix) and synthetic data, we randomly split the features. We reported detailed feature splitting for each task in the paper and included scripts in the supplement to reproduce the results. We will further test on data with complex structure and incorporate the results in the revision.
*Pathetic scenarios.* From our experimental studies, if Bob's data is not relevant (e.g., purely noisy or purposefully shuffled) to Alice's task, then Alice will observe unimproved or even degraded performance in the learning or/and prediction stages. If Bob is exactly the same as Alice, whether assisted learning will approach oracle depends on the underlying data and model. We will include more experiments and discussions on the pathetic scenarios in the revision.

[Meta-Review · NeurIPS 2020]

The paper proposes assisted learning, a framework for privacy-preserving learning for multiple organizations. The idea of sharing prediction residuals is very novel. The evaluations on medical applications are convincing.